# Segment-Specific Functional Responses of Swine Intestine to Time-Restricted Feeding Regime

**DOI:** 10.3390/ani16010052

**Published:** 2025-12-24

**Authors:** Hongyu Wang, Haoshu Shan, Xing Wei, Yong Su

**Affiliations:** 1College of Animal Science, Anhui Science and Technology University, Chuzhou 239000, China; 2019205029@njau.edu.cn; 2Laboratory of Gastrointestinal Microbiology, Jiangsu Key Laboratory of Gastrointestinal Nutrition and Animal Health, College of Animal Science and Technology, Nanjing Agricultural University, Nanjing 210095, China; 3Zhenjiang Animal Disease Prevention and Control Center, Zhenjiang 212000, China; 13405584208@163.com (H.S.); zx8883722@126.com (X.W.)

**Keywords:** growing pigs, time-restricted feeding, transcriptome, gene expression profiles

## Abstract

Evidence suggests that time-restricted feeding (TRF), where pigs are fed only during specific times, enhances growth performance, but its impact on different intestinal parts is unclear. In a study with 12 pigs, one group ate freely while the other had access to feed within specific times (7:00–8:00, 12:00–13:00, 17:00–18:00). After three weeks, we assessed nutrient digestion, enzyme activity, and gene expression in the jejunum and colon. Results showed that TRF improved protein digestion, altered enzyme activity, and varied gene expression, improving protein turnover in the jejunum and carbohydrate metabolism in the colon. These insights have the potential to enhance pig feeding strategies, thereby improving farm efficiency and promoting animal health.

## 1. Introduction

In animal husbandry, enhancing growth performance and feed utilization efficiency represents a fundamental objective, with the optimization of nutritional strategies being crucial to its attainment. Time-restricted feeding (TRF), an emerging feeding regimen, has been shown to significantly improve bodyweight gain and feed conversion efficiency in pigs without affecting total feed intake, indicating its potential for widespread application in large-scale swine production [1,2,3]. Emerging evidence suggests that TRF systematically influenced nutrient metabolism in pigs. Notably, TRF altered the hepatic transcriptome and metabolome, with differentially expressed genes (DEGs) and metabolites predominantly enriched in pathways related to fat and amino acid metabolism [1]. Additionally, TRF downregulates the expression of aromatic-L-amino-acid decarboxylase (*DDC*), tyrosine hydroxylase (*TH*), glutamic-oxaloacetic transaminase 2 (*GOT2*), and dopamine beta-hydroxylase (*DBH*), while increasing the relative concentrations of citrulline, kynurenine, L-tryptophan, and L-tyrosine in serum, all of which are associated with the metabolism of aromatic amino acids [2]. Furthermore, TRF has been shown to regulate metabolic health and alleviate chronic conditions such as obesity and insulin resistance in both human and rodent studies [3,4,5]. Its ability to impact systemic metabolism via the microbiota–gut–brain axis has garnered significant scholarly interest [6,7,8,9]. However, the precise intestinal mechanisms through which TRF enhances production performance in pigs remain insufficiently understood, particularly regarding the differential effects of TRF on the functions of various intestinal segments. This gap in knowledge presents a challenge to the optimization and implementation of TRF strategies within the framework of precision nutrition.

The intestine is key for nutrient digestion, absorption, and metabolism, with the small and large intestines having distinct roles. The small intestine mainly handles nutrient breakdown and absorption, whereas the large intestine, once thought to focus on water and electrolyte reabsorption, also ferments undigested carbohydrates, producing metabolites like short-chain fatty acids that influence integral metabolism and immunity. Research indicates that TRF changes nutrient substrates and microbial communities in the large intestine of pigs, potentially impacting its function significantly [10,11,12]. Most studies have currently focused on the impacts of TRF on regulating digestive and absorptive functions in the small intestine [13,14], with limited research on its effects on the large intestine and the differences in segment-specific responses. This gap hampers a full understanding of the role of TRF on intestinal function. Thus, a systematic comparison of the effects of TRF on the porcine small and large intestines and an analysis of segment-specific responses are essential to better understand TRF mechanisms.

Based on the aforementioned background, this study hypothesized that TRF enhances nutrient utilization in pigs by differentially affecting the small and large intestines. The feeding regimen employed in the current study closely resembled the three-meal pattern prevalent in contemporary society. Given the significant anatomical, physiological, polyphagic, habitual, metabolic, and gut microbiota similarities between pigs and humans, the findings from this study are expected to contribute valuable insights into the application of TRF both in swine production and humans. Thus, the study aimed to assess the impact of TRF on the digestive function of pigs and further to identify the gene expression profile and pathways specific to each intestinal segment via transcriptomics. These insights could optimize TRF strategies, improving swine production and health.

## 2. Materials and Methods

### 2.1. Ethics Statement

Nanjing Agricultural University Animal Care and Use Committee (Nanjing, China) (Approval#SYXK2019-0066) authorized the present research. All manipulations related to animals were carried out according to China Laboratory Animal Guideline for Ethical Review of Animal Welfare (GB/T 35892-2018) [15].

### 2.2. Animals, Housing, and Sampling

Twelve healthy growing crossbred (Duroc × Landrace × Yorkshire) pigs with an average bodyweight of 56.29 kg were employed in this study. Pigs were randomly (random number table method) divided into FA group and TRF group with each group having 6 replicates (1 pig/pen). The FA group had free access to feed, whereas the TRF pigs were fed ad libitum within 07:00–08:00, 12:00–13:00, and 17:00–18:00, respectively (Figure 1). The same commercial pellet feed was formulated according to the NRC (2012) [16]. Feed formulation is shown in Appendix A. All pigs were raised in individual pens under recommended feeding conditions. The experiment lasted for 3 weeks. All pigs were weighed and then euthanized by CO_2_ asphyxiation to collect the tissues, mucosa, and digesta samples from the digestive tract. The jejunum was harvested starting from the duodenojejunal flexure and ending at the jejunoileal junction. The entire colon was collected from the ileocecal valve to the distal end of the descending colon. After detaching the mesentery, the intestines were stretched to collect and homogenize the digesta for analysis. The length and weight of the jejunum and colon were measured respectively. A 20 cm middle section of the intestine was cut and cleaned with saline. Mucosal samples were collected by scraping the inner surface with a glass slide.

### 2.3. Tissue Indices of Jejunum and Colon

The tissue indices of the jejunum and colon were calculated using Formula (1):(1)$$Tissue\;index\left(\%\right)=\frac{weight\;of\;intestinal\;segment\times 100}{live\;body\;weight}$$

### 2.4. Apparent Total Tract Digestibility

To evaluate the apparent total tract digestibility (ATTD) of proximate nutrients, chromic oxide (Cr_2_O_3_) was utilized as an indigestible marker. During the final week of the trial, experimental pigs were administered diets supplemented with 0.3% Cr_2_O_3_. Fecal samples were collected over three consecutive days preceding the conclusion of the experiment. Fresh fecal matter was promptly stored at −20 °C to prevent degradation prior to analysis. Both feed and dried, finely ground fecal samples were analyzed in duplicate for their crude protein (CP), ether extract (EE), crude fiber (CF), crude ash (Ash), and chromium (Cr) content. All nutrient analyses adhered to official methods of AOAC (2000): Ash (method 920.153), CP (method 981.10), EE (method 922.06), and CF (Method 978.10). Nitrogen (N) content was determined using a Kjeldahl nitrogen analyzer (Foss8400, FOSS, Hillerød, Denmark), with CP calculated as N × 6.25. CF was analyzed utilizing an Ankom fiber analyzer (Ankom 200 fiber Analyzer, Ankom Technology, Macedon, NY, USA), and EE was measured using a Soxhlet extractor (Soxtec-2055, FOSS, Eden Prairie, MN, USA).

The concentration of Cr was quantified using a colorimetric method with a spectrophotometer (UV-1201, Shimadzu, Kyoto, Japan), in accordance with AOAC 974.25 and our previous method [17]. Samples underwent digestion in a mixture of concentrated nitric acid (HNO_3_) and perchloric acid (HClO_4_) on a hotplate within a fume hood. Digestion continued until dense white fumes emerged and the digestate exhibited a deep red coloration. Following cooling, the digestate was diluted to the desired volume with deionized water. The ATTD was calculated using Equation (2):(2)$$ATTD\;(\%)=[1-(Nutrient\_feed\times {Cr}_{2}O_{3}\_feces)/(Nutrient\_feces\times {Cr}_{2}O_{3}\_feed)]\times 100$$
where

Nutrient_feed = Concentration of the nutrient (CP, EE, CF, Ash) in the feed.

Nutrient_feces = Concentration of the nutrient in the feces.

Cr_2_O_3__feed = Concentration of Cr_2_O_3_ in the feed.

Cr_2_O_3__feces = Concentration of Cr_2_O_3_ in the feces.

### 2.5. Digestive Enzyme Activities

The analysis of digestive enzyme activities was conducted on stomach and ileum digesta, which were collected immediately following euthanasia. These samples were subsequently snap-frozen in liquid nitrogen. Samples were homogenized in PBS at a ratio of 1:9 (*w*/*v*) and centrifuged at 10,000× *g* for 15 min at 4 °C. Enzyme activities of pepsin (A080-1-1), chymotrypsin (A080-3-1), lipase (A054-2-1), amylase (C016-1-2), maltase (A082-3-1), and sucrase (A082-2-1) in the supernatants were quantified using commercial assay kits in accordance with the manufacturer’s protocols (Jiancheng Bioengineering Institute, Nanjing, China). Six samples per treatment were analyzed, with two technical repeats per each sample. The enzyme activities were analyzed and were expressed as units per gram of protein (U/g protein), with protein concentration determined using the Bradford method.

### 2.6. Colonic Nutrient Substrates

The concentrations of carbon (starch and cellulose) and nitrogen (total protein and NH3-N) in colonic digesta were measured [12]. Starch and cellulose were assessed using Solarbio Detection Kits BC0700 and BC4280 (Beijing, China), respectively. Total protein was measured with Solarbio Detection Kit PC0010 via the Coomassie brilliant blue G-250 method, whereas NH3-N levels were determined using a colorimetric method based on our previous work [12].

### 2.7. Total RNA Extraction and Transcriptome Analyses

Total RNA from jejunal and colonic mucosal samples were extracted with an RNA extraction kit (Tiangen BioTech Co., Ltd., Beijing, China). RNA sample concentration and integrity were assessed with a Nanodrop spectrophotometer (NanoDrop Technologies, Wilmington, NC, USA). RNA sequencing was conducted following cDNA library construction and sequencing on the Illumina HiSeq 2500 platform (Illumina, San Diego, CA, USA). The paired-end raw reads were submitted to the NCBI Sequence Read Archive (SRA) under accession number PRJNA1126326. Raw data were filtered to eliminate low-quality reads (Q20 ≤ 20), poly-N stretches (>5% N), and adaptor-only reads using WipeAdapter.pl (v.1.0.1) and Fastq (v.1.0.0) filter scripts [18]. Clean reads were then aligned to the pig reference genome (Sus scrofa 11.1) with TopHat (version 2.1.1), and gene expression levels were quantified using FPKM method. DEGs between the FA and TRF groups were identified using DESeq2, with criteria of |fold change| ≥ 1.5 and *p* value < 0.05 [19]. Gene ontology (GO) biological process and Kyoto Encyclopedia of Genes and Genomes (KEGG) enrichment and pathway analysis were conducted [20]. More details were described in our previously published articles [2].

### 2.8. Statistical Analysis

The statistical significance of differences in intestinal segment length, tissue indices, the ATTD of proximate nutrients, colonic nutrient substrates, and digestive enzyme activities between the two groups were calculated by *t*-test (two-tailed) (IBM SPSS Statistics for Windows, Version 21.0) and R software (version 4.1.0). Differences were considered significant at *p*  <  0.05, and a tendency was considered at 0.05 ≤ *p* < 0.1. Data visualization was performed with GraphPad Prism (version 8, GraphPad Software, San Diego, CA, USA), Cytoscape (Cytoscape-3.7.2), with built-in plugin ClueGO (version 2.5.8).

## 3. Results

### 3.1. Effects of TRF on the Length and Tissue Indices of Jejunum and the Colon of Pigs

In the present study, the feeding regime shift had no significant impact on tissue indices of either the jejunum (*p* = 0.80) or colon (*p* = 0.72) nor on the length of the colon (*p* = 0.19). However, the TRF pattern tended to enhance the length of the jejunum (*p* = 0.07, Table 1).

### 3.2. Effects of TRF on Apparent Total Tract Digestibility of Approximate Nutrients

Using the exogenous indicator method, we found that TRF significantly increased the ATTD of CP (increased by 18.9%, *p* = 0.045), whereas it exerted no significant effects on the ATTD of ash (*p* = 0.20), EE (*p* = 0.99) or CF (*p* = 0.71) compared with the FA group (Figure 2).

### 3.3. Effects of TRF on Digestive Enzyme Activities in the Gastro-Intestine of Pigs

Compared to the control group, TRF exerted no significant effect on pepsin activity (*p* = 0.13) in the stomach (Figure 3). In the ileum, TRF significantly decreased the activities of amylase (*p* < 0.01), sucrase (*p* < 0.01), and lipase (*p* < 0.01), whereas maltase (*p* = 0.12) and chymotrypsin (*p* = 0.33) activities remained unaffected (Figure 3). Furthermore, TRF significantly increased trypsin and lipase activities in the pancreas and increased amylase, maltase, and lipase activities in the duodenum; detailed data (Appendix A) for pancreatic and duodenal enzymes have been previously reported [2].

### 3.4. Effects of TRF on Colonic Nutrient Substrates of Pigs

Colonic nutrient substrates reflect, to some extent, the digestive efficiency in the small intestine and microbial metabolic activity in the large intestine. These substrates may influence systemic host metabolism via the microbiota–gut–brain axis. Our findings revealed that TRF significantly reduced colonic carbohydrate substrates, particularly starch (*p* = 0.02) and cellulose (*p* = 0.04) levels, whereas it exerted minimal effects on nitrogenous substrates total protein (*p* = 0.23) and NH3-N (*p* = 0.70) (Figure 4).

### 3.5. Jejunal Transcriptomic Responses of Growing Pigs Under Different Feeding Patterns

In the jejunum, a total of 55.93 Gb data were obtained, with an average of 6.54 Gb data per sample. The percentage of Q30 bases was greater than 92.69%, indicating reliable sequencing quality. These clean data were then mapped to the swine reference genome, resulting in mapping rates of 93.82%~94.52% (Appendix A). After splice prediction analysis, gene structure optimization analysis, and novel gene discovery based on the mapped genes, a total of 2033 novel genes were discovered. Among these novel genes, 1076 had functional annotations. In total, up to 13,493 genes were identified.

Principal component analysis (PCA) revealed a distinct transcriptomic pattern in response to the altered feeding pattern (Figure 5A). A total of 1339 DEGs were identified (Figure 5B). There were 462 genes up-regulated and 877 genes down-regulated. The DEGs with the greatest changes in expression were *NR1H4* (farnesoid X receptor, *p* = 9.61 × 10^−6^), *DDX58* (ATP-dependent RNA helicase DDX58, *p* = 8.57 × 10^−5^), *GPRL15* (G protein-coupled receptor 15, *p* < 0.001), *TMEM252* (transmembrane protein 252, *p* = 0.0042), *CUBN* (cubilin, *p* = 0.028), *NOL4* (nucleolar protein 4 like, *p* < 0.001), *SPTLC3* (serine palmitoyltransferase, *p* = 9.51 × 10^−5^), *GSDME* (gasdermin-E isoform X1, *p* = 8.26 × 10^−7^), *PKP1* (plakophilin 1, *p* = 3.97 × 10^−4^), *TMIGD1* (transmembrane and immunoglobulin domain-containing protein 1, *p* = 9.66 × 10^−5^), *FABP6* (fatty acid-binding protein 6, *p* = 2.50 × 10^−7^), and *SLC10A2* (solute carrier family 10 member 2, *p* = 2.81 × 10^−7^). DEGs were mainly associated with 215 GO terms (Figure 5C).

The most enriched biological process GO terms were cell migration and biological adhesion. The most enriched molecular function GO terms were cell adhesion molecule binding, protein tyrosine kinase activity, serine hydrolase activity, serine-type peptidase activity, endopeptidase activity, metalloendopeptidase activity, and collagen binding. KEGG analysis suggested that these DEGs were mainly enriched in 43 KEGG pathways. The most enriched KEGG pathways (Figure 5D) were focal adhesion, ECM–receptor interaction, protein digestion and absorption, complement and coagulation cascades, and axon guidance. Of note, *SLC1A1* (Solute Carrier Family 1 Member 1, *p* = 0.003), *SLC38A2* (Solute carrier family 38 member 2, *p* = 0.047), *SLC7A8* (Solute carrier family 7 member 8, *p* = 0.049), *SLC8A1* (Solute carrier family 8 member 1, *p* = 0.014), and *XPNPEP2* (X—Prolyl aminopeptidase 2, *p* = 0.023), which are closely related to the digestion and absorption of protein, were significantly altered by the TRF regime.

### 3.6. Colonic Transcriptomic Responses of Growing Pigs Under Different Feeding Patterns

In the colon, 59.16 Gb data were obtained after quality control, with an average of 6.80 Gb data per sample. The percentage of Q30 bases was 92.73%. The mapping rates ranged from 93.57% to 94.69%. A total of 2291 novel genes (1130 with functional annotations) were identified (Appendix A). The PCA results of the TRF and FA groups failed to clearly distinguish the two groups (Figure 6A). From the volcano plot (Figure 6B), among these 13,858 genes, there were 268 mRNAs which were significantly altered (175 up-regulated and 93 down-regulated). The most significantly down-regulated DEGs with maximal fold change were *FAM53C* (family with sequence similarity 53 member C, *p* = 2.18 × 10^−3^), *G6PC* (glucose-6-phosphatase, *p* = 0.031), *DMGDH* (dimethylglycine dehydrogenase, *p* = 0.081), and *PGLYRP2* (peptidoglycan recognition protein, *p* = 0.005), whereas the most significantly up-regulated DEGs were *LYNX1* (Ly6/neurotoxin 1, *p* = 0.001), *EFCAB7* (EF-hand calcium-binding domain-containing protein 7, *p* = 0.013), *PKP1* (Plakophilin 1, *p* = 0.034), and *DCAF5* (DDB1-and CUL4-associated factor 5, *p* = 2.06 × 10^−6^). Further, the GO pathway enrichment plot (Figure 6C) revealed that the altered genes were mainly related to 127 GO terms.

The most enriched biological process GO terms were microtubule-based movement regulation, regulation of cilium movement, heart development, multicellular organismal reproductive process, and monosaccharide transmembrane transport. The most enriched cell components GO terms were the intrinsic component of plasma membrane, sperm flagellum, 9 + 2 motile cilium, postsynaptic specialization, and extracellular organelle. The most enriched molecular functions GO terms were glycosaminoglycan binding, signaling adaptor activity, CCR chemokine receptor binding, cadherin binding, protein tyrosine kinase binding, and transferring pentosyl groups. KEGG analysis revealed that the DEGs were mainly related to starch and sucrose metabolism, pyrimidine metabolism, axon guidance, pertussis, gap junction, and circadian entrainment plot (Figure 6D).

### 3.7. Comparative Transcriptomic Profiles Between Jejunum and the Colon of Growing Pigs Under Different Feeding Patterns

Comparatively, five times more DEGs were identified in the jejunum than in the colon (1339 vs. 268, Figure 7A). Of these genes, only 62 DEGs were shared between the jejunum and the colon. These genes were mainly enriched in KEGG pathways related to axon guidance, which might implicate the bidirectional interaction between the gut and the brain [21]. In the jejunum, 462 genes were up-regulated and 877 were down-regulated, whereas in the colon, 175 genes were up-regulated and 93 genes were down-regulated (Figure 7B). The significantly changed genes were primarily related to KEGG pathways related to physiological processes, pathological processes, and nutrient metabolism (Figure 7C). Notably, DEGs in the jejunum were mainly related to the digestion and absorption of protein, whereas DEGs in the colon were primarily related to pyrimidine metabolism and starch and sucrose metabolism pathways.

## 4. Discussion

Studies showed that TRF helps reduce weight and improve metabolic health in humans and mice, while in pigs, it enhances production performance and weight gain [5,13,14]. This discrepancy arises because TRF in humans typically focuses on weight and metabolic management, often involving reduced dietary energy intake, whereas in pigs, the goal is to boost nutrient intake and utilization efficiency. Despite these differences, both fields illustrate that TRF effectively regulates nutrient metabolism. Our previous research revealed that TRF improved the bodyweight and feed efficiency in pigs without changing the total feed intake [1]. In the current study, we observed that TRF tended to increase the jejunum length and significantly enhanced the ATTD of CP. Further analyses revealed elevated activities of proteases, lipases, and amylases in the pancreas and duodenum but decreased digestive enzyme activities in the ileum. These findings collectively illustrate a compensatory growth response and metabolic reprogramming of the porcine intestine induced by TRF. In the duodenum, which serves as the primary site for nutrient digestion, enzymatic activity is predominant. In contrast, the ileum, functionally specialized for nutrient absorption, demonstrates an adaptive down-regulation of enzyme secretion. This physiological adaptation likely represents an energy conservation strategy, reducing unnecessary enzymatic production for substrates already digested proximally.

Transcriptomic analysis identified DEGs involved in pathways concerning immune function, disease, and nutrient metabolism, which can further affect various aspects of gastrointestinal function, such as nutrient absorption, barrier integrity, and immune responses [22,23]. Regarding nutrient metabolism, DEGs in the small intestine were enriched in pathways related to protein digestion and absorption, with up-regulated expression of genes encoding amino acid transporters (e.g., *SLC1A1*, *SLC38A2*) and peptidases (e.g., *XPNPEP2*). Simultaneously, changes in the expression of genes involved in protein digestion and absorption and lipid metabolism pathways were also detected in the liver and hypothalamus [1,2]. Furthermore, transcriptomic analysis revealed an up-regulatedd expression of the *FXR* in the jejunum, a critical bile acid-sensing nuclear receptor. Under TRF conditions, the proximal small intestine, specifically the jejunum, demonstrates enhanced nutrient digestion, as evidenced by increased pancreatic chymotrypsin and lipase activity. This enhancement subsequently facilitates bile acid reabsorption through the *SLC10A2* in the jejunum. The mechanisms by which digestive enzyme release is regulated warrant further investigation. It should be noted that the concentration of bile acids was not measured in this study, which presents an opportunity for future research. In addition, only transcriptomics analysis was performed in the present study. Further validation can be conducted via methods such as Western blotting or proteomics in future research.

Indeed, TRF induced differential impacts on the gene expression profiles of the large and small intestines, aligning with the distinct functional roles of these gastrointestinal segments. The small intestine is mainly responsible for nutrient digestion and absorption. In this context, it is notable that under TRF, DEGs in the small intestine were primarily related to protein digestion and absorption pathways and fat metabolism. A recent study found that compared to a single feeding, groups fed three and five times a day showed significant improvements in CP digestibility and in the activities of trypsin and chymotrypsin [13]. Alterations in feeding times influence the dynamic changes in nutrient and hormone levels in serum [24]. These findings suggest that TRF may particularly influence the way the small intestine processes and assimilates proteins, possibly by optimizing protein metabolism and utilization. These differences might be due to variations in TRF patterns, which can have distinct effects on outcomes. For example, a 2-meals-per-day TRF regimen showed reduced lipid accumulation and improved inflammatory responses compared to a 12-meals-per-day TRF [14]. An 8 h TRF resulted in lower energy intake and higher levels of ghrelin and adiponectin compared to a 12 h TRF regimen [25]. Feeding during the light phase within a 12 h TRF window led to a 13% greater weight gain [26].

Traditionally, the main function of the large intestine was thought to be electrolyte and water reabsorption. It is now established that undigested carbohydrates can be fermented by microbiota in the large intestine [27]. TRF significantly lowered colon levels of carbon substrates like starch and cellulose but had a lesser effect on nitrogenous substrates. This is likely because, although TRF boosts chymotrypsin activity, efficient protein digestion and absorption in the small intestine result in similar levels of nitrogen reaching the colon. Thus, colonic nitrogen mainly originates from endogenous and microbial sources. This aligns with high protein digestibility, as increased microbial biomass under TRF may use this nitrogen to produce more microbial protein or bioactive compounds, enhancing the the ATTD of CP in pigs. This observation prompts the question regarding why reduced carbohydrate levels did not significantly affect the ATTD of CF. Although there was an 18% numerical increase in digestibility, high individual variability prevented statistical significance. Additionally, methodological differences are also important: CF includes cellulose, lignin, and hemicellulose and is measured differently from colonic cellulose. Colonic substrates are measured using specific assay kits, whereas CF relies on proximate analysis. In the proximate nutrient analysis protocol, the CF fraction contains not only cellulose but also portions of lignin and hemicellulose, whereas the cellulose assay kit only determines the cellulose fraction.

Along with these changes in carbohydrate substrates, DEGs in the large intestine under TRF are mainly concentrated in pathways associated with starch and sucrose metabolism, indicating a shift in how this segment handles complex carbohydrates. This could reflect changes in microbial configuration and activity in the large intestine, as these microbes play a crucial role in breaking down starches and sugars not digested in the small intestine [28,29]. Our previous results found that TRF reduced the fiber concentration in the colon and altered its dynamic fluctuations [12]. Also, TRF affects the gut microbiome composition, diversity, and microbial metabolites [11,30]. Interestingly, the fluctuations in nutrients drive dynamic changes in the microbiome. In turn, these microbial communities have implications for host metabolism, immune system, and behavior [31,32].

These DEG patterns highlight the nuanced and segment-specific responses of the gastrointestinal tract to dietary interventions such as TRF. It also entrains peripheral circadian clocks in the gut, impacting hormonal secretion, liver gene expression, and nutrient metabolism [33,34]. Shifting feeding time, whether postponed or brought forward, can lead to reversible changes in body temperature and the overall activities of juvenile mice [35]. Extended periods of restricted feeding times have the potential to disrupt rhythmic patterns of blood glucose, triglyceride, and high-density lipoprotein levels [36]. TRF has been found to induce circadian phase shifts in various physiological markers, independent of the hypothalamic suprachiasmatic nucleus. Additionally, TRF significantly influenced the diurnal patterns of serum cytokines. Moreover, meal timing may have implications for bodyweight regulation [37]. Consuming meals in the evening may lead to a misalignment between the central and peripheral biological clocks, potentially impacting the gut microbiota and gastrointestinal function [38]. As evidence, DEGs shared between the jejunum and colon were primarily enriched in the axon guidance pathway which might implicate the gut–brain interaction. Research has shown that TRF changes amino acid and lipid metabolism without directly impacting core clock gene expression [39]. Additionally, it restored disrupted intestinal rhythmicity, suggesting that its effects on the circadian rhythms may be mediated by the gut microbiome [40]. This suggests a significant interplay between dietary patterns, metabolic processes, and gut microbiota, emphasizing the importance of feeding time relative to circadian biology. The restoration of gut rhythmicity through TRF has potential therapeutic implications for conditions associated with circadian disruptions and metabolic disorders. These findings underscore the adaptability of the gut to changes in feeding patterns, with each segment responding in a way that supports its unique role in digestion and nutrient absorption. Understanding these changes is crucial for comprehending the full scope of the impact of TRF on gastrointestinal function and overall metabolic health.

## 5. Conclusions

In conclusion, our research indicated that TRF preferentially enhanced protein assimilation in the proximal intestine and carbohydrate metabolism in the distal intestine, likely facilitated by microbial fermentation. These variations in the gene expression profiles of different intestinal segments induced by TRF were adaptive to intestinal functions. These findings offer mechanistic insights for optimizing TRF strategies within the context of precision nutrition.

## Figures and Tables

**Figure 1 animals-16-00052-f001:**
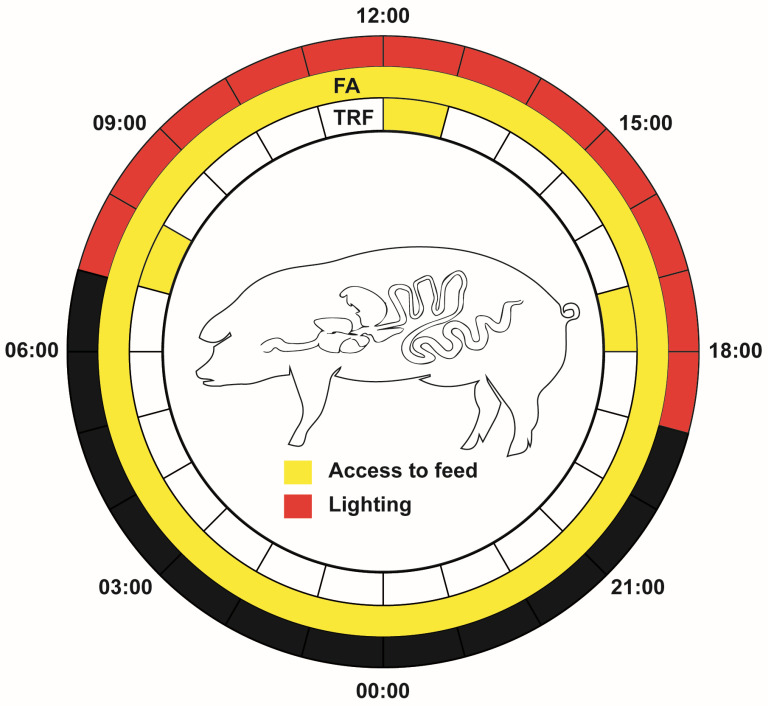
Flowchart of the experiment. FA: free access feeding; TRF: time-restricted feeding.

**Figure 2 animals-16-00052-f002:**
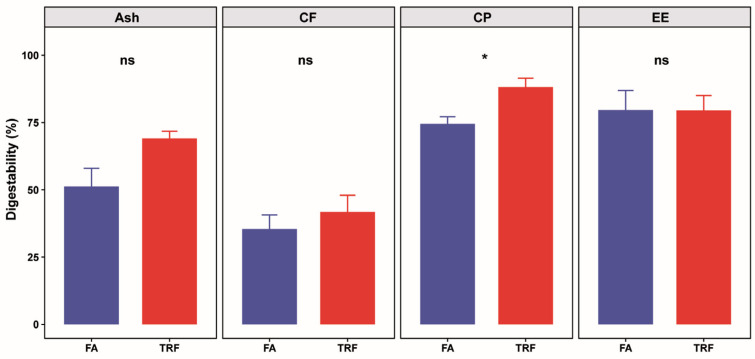
Effects of TRF on apparent total tract digestibility (ATTD) of proximate nutrients (n = 6). CF: crude fiber, CP: crude protein, EE: ether extract. ns denotes non-significant difference (*p* > 0.05) between the FA group and the TRF group whereas * denotes a significant difference (*p* < 0.05).

**Figure 3 animals-16-00052-f003:**
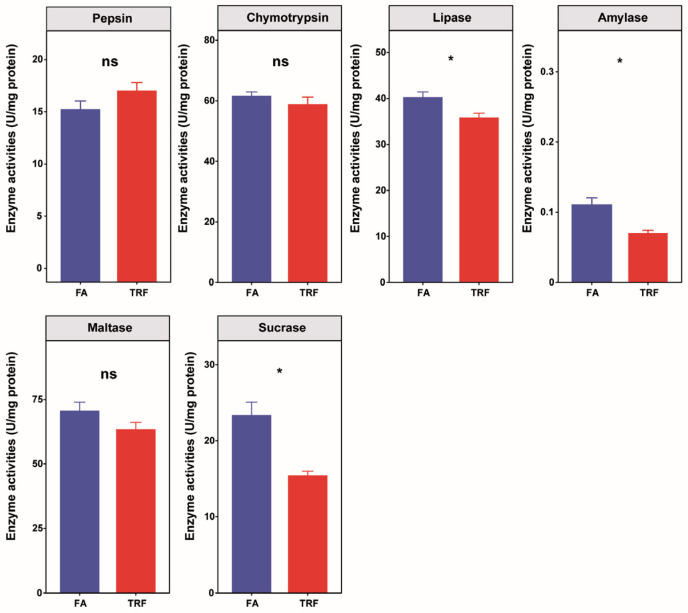
Effects of TRF on digestive enzyme activities within the stomach and the ileum of growing pigs (n = 6). ns denotes non-significant difference (*p* > 0.05) between the FA group and the TRF group whereas * denotes a significant difference (*p* < 0.05).

**Figure 4 animals-16-00052-f004:**
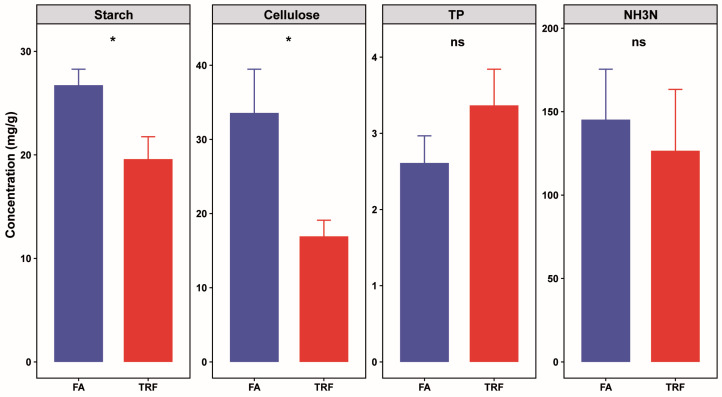
Effects of TRF on nutrient substrates in the colonic digesta of growing pigs (n = 6). TP = total protein. ns denotes non-significant difference (*p* > 0.05) between the FA group and the TRF group whereas * denotes a significant difference (*p* < 0.05).

**Figure 5 animals-16-00052-f005:**
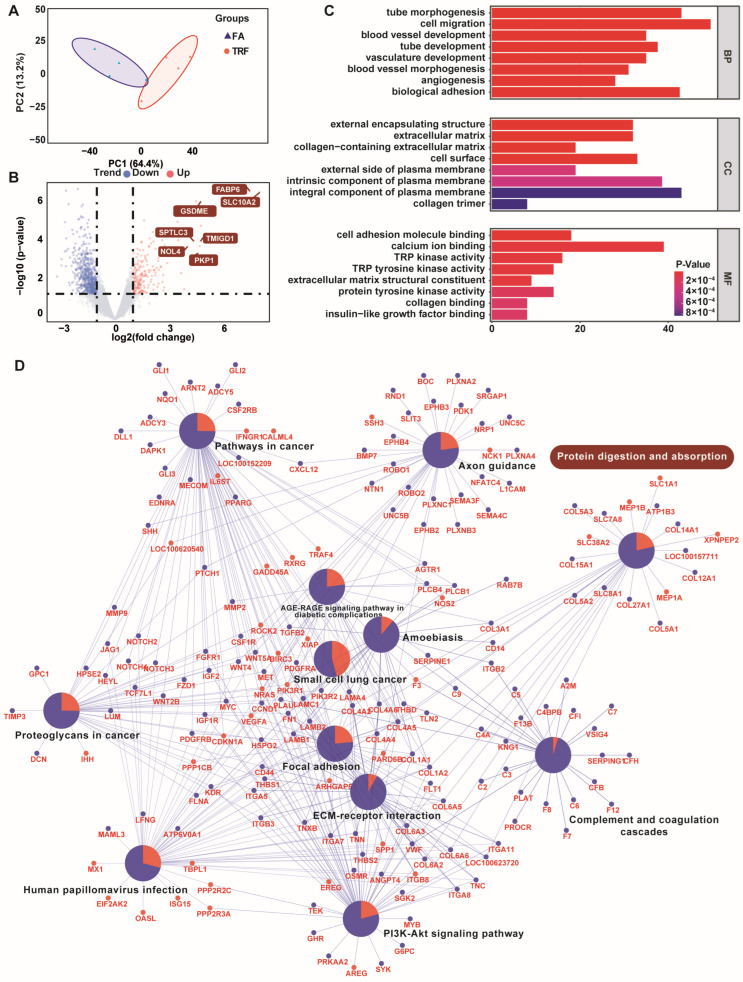
Summary of the transcriptome analyses in the jejunum of growing pigs under different feeding modes (n = 4). (**A**) PCA between the FA group and TRF group. (**B**) Volcano plot exhibiting the distributions of the DEGs between the FA group and TRF group. (**C**) Bar plot showing the enriched GO terms of DEGs. BP: biological process. CC: cellular component. MF: molecular function. (**D**) KEGG pathway enrichment analysis exhibiting DEG-enriched pathways. The central node with pie chart presents enriched pathways whereas radiated nodes around them exhibit enriched DEGs. DEGs in red color were up-regulated whereas blue-colored DEGs were down-regulated. Pie charts represent the percentage of up-regulated and down-regulated DEGs within the pathway.

**Figure 6 animals-16-00052-f006:**
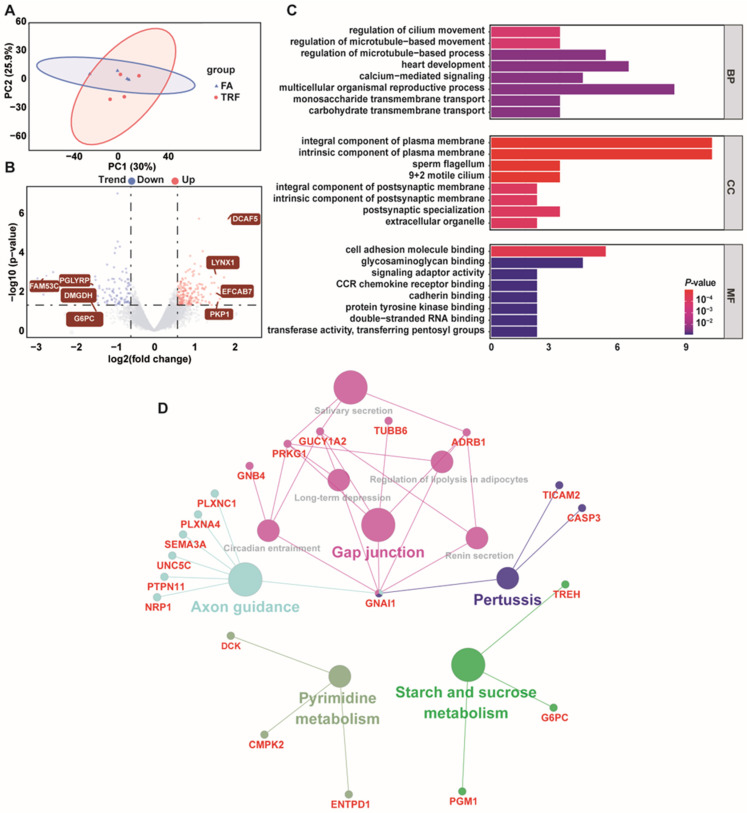
Summary of the transcriptome analyses in the colon of growing pigs under different feeding modes (n = 4). (**A**) PCA between the FA group and TRF group. (**B**) Volcano plot exhibiting the distributions of the DEGs between the FA group and TRF group. (**C**) Bar plot showing the enriched GO terms of DEGs. BP: biological process. CC: cellular component. MF: molecular function. (**D**) KEGG pathway enrichment analysis exhibiting DEG-enriched pathways. Colored nodes present enriched pathways whereas radiated nodes around them exhibit enriched DEGs.

**Figure 7 animals-16-00052-f007:**
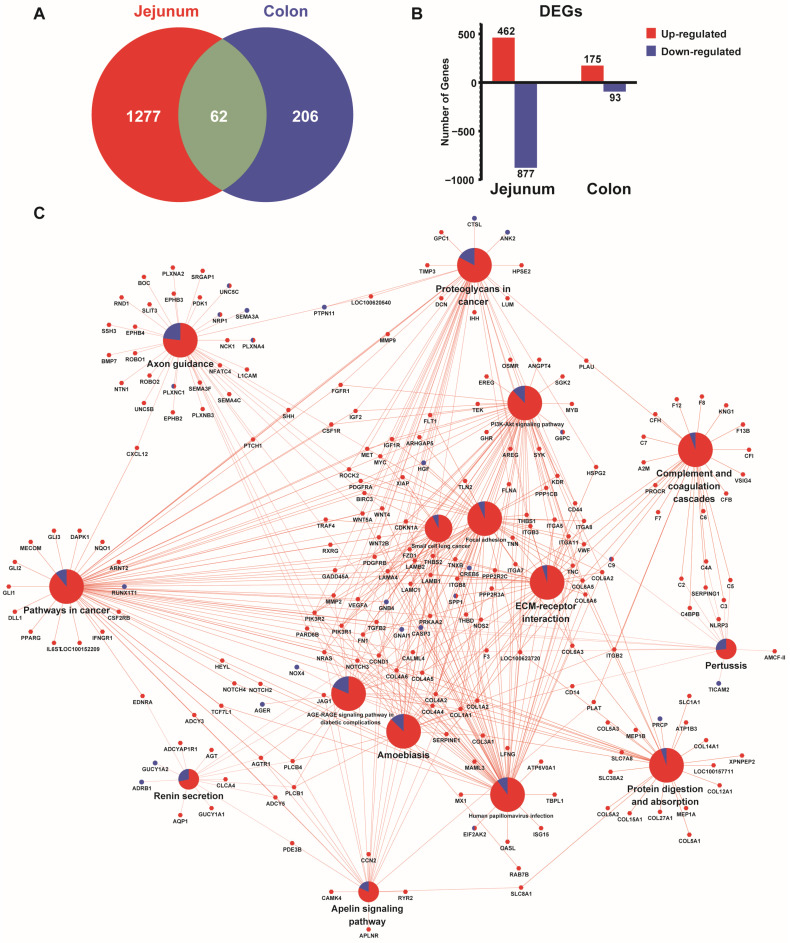
Comparative transcriptomic profiles between different intestinal segments of growing pigs under different feeding patterns (n = 4). (**A**) Venn diagram revealing the numbers of DEGs identified in different intestinal segments. (**B**) Column chart depicting the number of DEGs up-regulated and down-regulated in different intestinal segments. (**C**) KEGG pathway enrichment analysis exhibiting DEG-enriched pathways specific to different intestinal segments. The central node with pie chart represents co-enriched pathways of DEGs in both jejunum and colon, whereas node size represents the number of DEGs enriched in the pathway. Radiated nodes around them exhibit enriched pathways. DEGs in red were from the jejunum whereas blue-colored DEGs were from the colon. The pie chart represents the ratio of jejunal to colonic DEGs contributing to that pathway.

**Table 1 animals-16-00052-t001:** Effects of TRF on swine intestinal length and organ indices (n = 6).

Index	Segment	Treatment		*p* Value
		FA	TRF	
Length (m)	Jejunum	17.76 ± 0.85	19.10 ± 0.41	0.067
	Colon	3.97 ± 0.21	4.32 ± 0.15	0.188
Organ index (%)	Jejunum	2.78 ± 0.14	2.84 ± 0.18	0.803
	Colon	2.57 ± 0.15	2.71 ± 0.33	0.721

FA = free access; TRF = time-restricted feeding. Data are presented as mean ± SEM.

## Data Availability

The RNA sequencing data related to the present manuscript has been submitted to the Sequence Read Archive (SRA) database (https://www.ncbi.nlm.nih.gov/sra, accessed on 20 December 2025) with an accession number of PRJNA1126326.

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
