# Peer review of "Segment-Specific Functional Responses of Swine Intestine to Time-Restricted Feeding Regime"

_animals, 2025, doi:10.3390/ani16010052_

Round 1

Reviewer 1 Report

Comments and Suggestions for Authors

This paper studied the specific effects of time-restricted feeding (TRF) on different segments of the swine intestine, highlighting how TRF differentially regulates protein absorption in the jejunum and carbohydrate metabolism in the colon. It addresses a research gap in understanding the segment-specific impacts on swine intestinal function of TRF and supports optimizing feeding strategies in precision nutrition. Overall, the paper is well-designed with significant theoretical value and application prospects. However, it requires substantial revisions before acceptance. Specific suggestions are as follows:

  1. The paper only mentions collecting tissues from the middle sections of the jejunum and colon but fails to clearly define their specific anatomical scope (e.g., the boundaries between the jejunum and duodenum/ileum, and the specific subsegments of the colon). Differences in intestinal segment division among researchers may affect the reproducibility of experimental results. It is recommended to supplement the clear anatomical definitions of intestinal segments and standardized operating procedures for sampling locations (e.g., specific distances from the pylorus/cecum) in the "Materials and Methods" section.
  2. The paper states that TRF affects digestive enzyme activities in the pancreas and duodenum, but only presents enzyme activity data of the stomach and ileum in the results. Detailed data on the pancreas and duodenum are merely cited from previous research [2] without supplementary presentation in this paper. Considering that the pancreas and duodenum are key sites for nutrient digestion, changes in their enzyme activities are crucial for understanding the regulatory mechanism of TRF. It is recommended to supplement relevant detailed data in the supplementary materials to ensure the integrity of the study.
  3. In discussion section does not explicitly mention the limitations of this study nor propose specific future research directions. It is recommended to add an analysis of research limitations at the end of the discussion and put forward targeted future research plans.
  4. Why was the TRF pattern chosen? This should be clarified in the paper.
  5. The ordinate labels of Figure 3 (digestive enzyme activities) are unclear (e.g., some coordinate values are missing, and units are not specified). It is recommended to correct the ordinate labels, supplement complete unit information (e.g., "U·g⁻¹ protein"), and ensure the readability of the graph data.
  6. Language issues: In "Our precious research revealed that TRF improved..." (Discussion section, Line 301), "precious" is a semantic misuse and should be replaced with "previous" (indicating "prior research" rather than "valuable").
  7. Spelling error: In "statistical significance of differences in intestinal segment length, tissue induces..." (Section 2.8, Line 157), "tissue induces" is incorrect. The correct plural form of "index" is "indices", while "induces" is a verb meaning "to induce". Thus, it should be revised to "tissue indices".
  8. Reference 1: "Xia, P.-k.; Wang, H.-y.; Zhang, H.; Su, Y.; Zhu, W.-y. Effects of time-restricted feeding on the growth performance and liver metabolism of pigs. 2022, 53, 2228-2238" (Line 430) lacks the journal name.

Author Response

This paper studied the specific effects of time-restricted feeding (TRF) on different segments of the swine intestine, highlighting how TRF differentially regulates protein absorption in the jejunum and carbohydrate metabolism in the colon. It addresses a research gap in understanding the segment-specific impacts on swine intestinal function of TRF and supports optimizing feeding strategies in precision nutrition. Overall, the paper is well-designed with significant theoretical value and application prospects. However, it requires substantial revisions before acceptance. Specific suggestions are as follows:

  1. The paper only mentions collecting tissues from the middle sections of the jejunum and colon but fails to clearly define their specific anatomical scope (e.g., the boundaries between the jejunum and duodenum/ileum, and the specific subsegments of the colon). Differences in intestinal segment division among researchers may affect the reproducibility of experimental results. It is recommended to supplement the clear anatomical definitions of intestinal segments and standardized operating procedures for sampling locations (e.g., specific distances from the pylorus/cecum) in the "Materials and Methods" section.

Answer: Thank you for your valuable suggestions; we have supplemented the details in the Materials and Methods section.

  1. The paper states that TRF affects digestive enzyme activities in the pancreas and duodenum, but only presents enzyme activity data of the stomach and ileum in the results. Detailed data on the pancreas and duodenum are merely cited from previous research [2] without supplementary presentation in this paper. Considering that the pancreas and duodenum are key sites for nutrient digestion, changes in their enzyme activities are crucial for understanding the regulatory mechanism of TRF. It is recommended to supplement relevant detailed data in the supplementary materials to ensure the integrity of the study.

Answer: Thank you for your valuable suggestions; we have supplemented the data referring to digestive enzyme activities in the pancreas and duodenum in the Figure S1 and cited in the manuscript.

  1. In discussion section does not explicitly mention the limitations of this study nor propose specific future research directions. It is recommended to add an analysis of research limitations at the end of the discussion and put forward targeted future research plans.

Answer: We appreciate your suggestions. In the Discussion section, we have added a discussion on the limitations (methodology and etc.) of this paper, and further proposed future research directions addressing these limitations.

  1. Why was the TRF pattern chosen? This should be clarified in the paper.

Answer: Thanks for your question. The feeding regimen employed in the current study closely resembled the three-meal pattern prevalent in contemporary society. Given the significant anatomical, physiological, polyphagic, habitual, metabolic, and gut microbiota similarities between pigs and humans, the findings from this study are expected to contribute valuable insights into the application of TRF both in swine production and humans.

  1. The ordinate labels of Figure 3 (digestive enzyme activities) are unclear (e.g., some coordinate values are missing, and units are not specified). It is recommended to correct the ordinate labels, supplement complete unit information (e.g., "U·g⁻¹ protein"), and ensure the readability of the graph data.

Answer: We appreciate your suggestion. We have improved the coordinates and units in the Figure 3 and replaced the original one in the manuscript.

  1. Language issues: In "Our precious research revealed that TRF improved..." (Discussion section, Line 301), "precious" is a semantic misuse and should be replaced with "previous" (indicating "prior research" rather than "valuable").

Answer:We appreciate your valuable comments. We have fully checked the grammatical and spelling issues throughout the manuscript to improve the readability of the paper.

  1. Spelling error: In "statistical significance of differences in intestinal segment length, tissue induces..." (Section 2.8, Line 157), "tissue induces" is incorrect. The correct plural form of "index" is "indices", while "induces" is a verb meaning "to induce". Thus, it should be revised to "tissue indices".

Answer: We appreciate your valuable comments. We have fully checked the grammatical and spelling issues throughout the manuscript to improve the readability of the paper.

  1. Reference 1: "Xia, P.-k.; Wang, H.-y.; Zhang, H.; Su, Y.; Zhu, W.-y. Effects of time-restricted feeding on the growth performance and liver metabolism of pigs. 2022, 53, 2228-2238" (Line 430) lacks the journal name.

Answer: We appreciate your valuable comments. We have double checked reference list and revised in the manuscript.

Reviewer 2 Report

Comments and Suggestions for Authors

Dear authors, the manuscript is well written but should be revised in some places. The sections to be updated are specified below.

-Abstract and Simple Summary should be more concise, limiting themselves to the main results and their implications.

-The introduction should include additional bibliographical references, expanding on the concepts described.

-The Discussion should relate directly to the data on pigs presented in the study.

-Figures and tables should include clearer captions and indicate the number of biological replicates.

The study is scientifically interesting, however, major revisions are necessary to enhance scientific rigor, organization, and clarity before the manuscript can be considered for publication.

Comments on the Quality of English Language

A linguistic  revision is recommended to bring it into line with Animals standards.

Author Response

Dear authors, the manuscript is well written but should be revised in some places. The sections to be updated are specified below. The study is scientifically interesting, however, major revisions are necessary to enhance scientific rigor, organization, and clarity before the manuscript can be considered for publication. 

  1. Abstract and Simple Summary should be more concise, limiting themselves to the main results and their implications.

Answer: We appreciate your suggestion and have revised and restructured the Abstract and Simple Summary sections to comply with the guidelines of Animals.

  1. The introduction should include additional bibliographical references, expanding on the concepts described.

Answer: Thanks for your suggestion. We have supplemented relevant literature in the Introduction section.

  1. The Discussion should relate directly to the data on pigs presented in the study.

Answer: We appreciate your suggestion. In the Discussion section, we integrated other experimental data from the same batch of animal trials which allows readers to gain a holistic understanding of indicators such as production performance throughout the entire experiment, and in this context, they can better comprehend certain indicators reported in the present study.

  1. Figures and tables should include clearer captions and indicate the number of biological replicates.

Answer: Thanks for your suggestion. We have optimized the captions of each figure and table, with the addition of information regarding biological replicates.

  1. A linguistic revision is recommended to bring it into line with Animals standards.

Answer: We appreciate your valuable comments. We have fully checked the linguistic issues throughout the manuscript to improve the readability of the paper.

Reviewer 3 Report

Comments and Suggestions for Authors

Dear authors your manuscript entitled “Segment-specific functional responses of swine intestine to 2 time-restricted feeding regime” aims to investigate the functional role of different intestine segments (jejunum and and colon) in response to the time restricted feeding regime (TRF). Effectively a wide amount of published papers in recent years reports on the benenifical effects of TRF on health and on metabolism (studies conducted in human and animals) also evaluating the role of immune system without consider the role of intestine. The reaserch project is interesting and well conceived. However in the complex the manuscript requires minor revisions.

The authors should precise some more informations about the animals groups because an identical number of pigs are cited in a previous published paper (Time-restricted feeding affects colonic nutrient substrates and modulates the diurnal fluctuation of microbiota in pigs. Front. Microbiol. 14:1162482.doi: 10.3389/fmicb.2023.1162482s).Identical is also the ethics approval.

The authors should explain if the pigs used for tjhis study belongs to the same group of previous published paper and in the case cite the article in the materials and methods. Moreover in the section 2.5 the authors should descrive better the enzymes evaluated and more detailed information on the assay hkit used (code number for each enzyme tested). In addition, the authors should add howmany determinations and replicates were performed for each enzyme and sample (pig). Moreover, the authors should indicate the coefficient of inter and intra variability among the values determined amongs the animals in the groups.

Some minor revisions are regarding the reference list. It is necessary to introduce the Journal (name, volume and pages) for reference 1. Moreover, the Journal titles have to be revised acoording the style requested in the author’s instructions of the Journal.  In addition, it is necessary to complete the references n. 15, 17 and 34.

Author Response

Dear authors your manuscript entitled “Segment-specific functional responses of swine intestine to 2 time-restricted feeding regime” aims to investigate the functional role of different intestine segments (jejunum and colon) in response to the time restricted feeding regime (TRF). Effectively a wide amount of published papers in recent years reports on the beneficial effects of TRF on health and on metabolism (studies conducted in human and animals) also evaluating the role of immune system without consider the role of intestine. The research project is interesting and well conceived. However in the complex the manuscript requires minor revisions.

  1. The authors should precise some more information about the animals groups because an identical number of pigs are cited in a previous published paper (Time-restricted feeding affects colonic nutrient substrates and modulates the diurnal fluctuation of microbiota in pigs. Front. Microbiol. 14:1162482.doi: 10.3389/fmicb.2023.1162482s).Identical is also the ethics approval.

Answer: We appreciate your attention and carefulness. The reference mentioned and this paper are, in fact, derived from the same animal experiment, but they focus on different research aspects and differ in the timing of sample collection. In addition, due to fistula issues in the animals involved in the references, one pig was excluded from each treatment group, whereas in this study, samples were collected from all 12 pigs.

  1. The authors should explain if the pigs used for this study belongs to the same group of previous published paper and in the case cite the article in the materials and methods. Moreover,in the section 2.5 the authors should describe better the enzymes evaluated and more detailed information on the assay kit used (code number for each enzyme tested). In addition, the authors should add how many determinations and replicates were performed for each enzyme and sample (pig). Moreover, the authors should indicate the coefficient of inter and intra variability among the values determined among the animals in the groups. 

Answer: Thanks for your question. As mentioned in Q1, the reference mentioned and this paper are derived from the same animal experiment, but they focus on different research aspects and differ in the timing of sample collection. In section 2.5, we have supplemented more experimental details, including the specific information of the assay kits and the detailed operational procedures. 

  1. Some minor revisions are regarding the reference list. It is necessary to introduce the Journal (name, volume and pages) for reference 1. Moreover, the Journal titles have to be revised acroding the style requested in the author’s instructions of the Journal.  In addition, it is necessary to complete the references n. 15, 17 and 34.

Answer: We appreciate your valuable comments. We have double checked reference list and revised in the manuscript.

Reviewer 4 Report

Comments and Suggestions for Authors

Based on the evaluation of the submitted manuscript titled "Segment-specific functional responses of swine intestine to time-restricted feeding regime," this work addresses a relevant and timely topic in swine production by examining the effects of time-restricted feeding on intestinal function. The discovery that time-restricted feeding affects the jejunum and colon differently, improving protein utilization in the upper intestine and carbohydrate metabolism in the lower intestine, is novel and makes an important contribution to the field. The methods are appropriate overall, and the results are clearly presented. However, some points would benefit from further clarification and slight expansion to enhance the manuscript’s conclusions and overall significance. Please kindly follow the comments and suggestions provided below;

  1. The study indicates elevated pancreatic chymotrypsin and lipase activities alongside reduced amylase, sucrase, and lipase activities in the ileum (Lines 191-196). The authors suggest this pattern reflects an energy conservation approach, given more proximal digestion of substrates (Lines 314-318). Although this explanation seems reasonable, it represents a key interpretation that would gain from additional evidence. Could the authors elaborate on potential signaling pathways underlying this regional enzyme reduction? For example, might changes in nutrient or bile acid flow from time-restricted feeding activate feedback loops that limit enzyme production in the distal small intestine?
  2. The manuscript reports a notable 18.9% rise in Apparent Total Tract Digestibility (ATTD) of crude protein (Lines 185-188). Authors then link this to potential boosts in colonic microbial protein synthesis, drawing on nitrogen from endogenous and microbial pools (Lines 352-354). This hypothesis holds interest, yet ATTD for crude protein incorporates microbial nitrogen, which may complicate assessments of actual protein uptake. Did the authors evaluate apparent ileal digestibility (AID) of amino acids? Such analysis would offer a sharper gauge of small intestine protein absorption, separating it from hindgut fermentation influences and bolstering claims that time-restricted feeding improves proximal protein use.
  3. The study observes that time-restricted feeding (TRF) lowered colonic cellulose by 18% significantly, yet showed no notable change in Apparent Total Tract Digestibility (ATTD) of crude fiber (CF) (Lines 203-206). The authors appropriately highlight high individual variation and differences in CF versus cellulose measurement methods (Lines 354-359). This discussion merits further development, as the cellulose drop represents a core result tied to shifts in hindgut metabolism, while the absent ATTD-CF effect needs deeper exploration. Could components within CF, such as unmeasured lignin, account for this inconsistency, since cellulose assays exclude it? Addressing this would enrich the interpretation with greater detail.
  4. The study reports a trend toward increased jejunum length in the time-restricted feeding (TRF) group (P=0.07, Line 177), which, although not statistically significant, is a notable morphological observation. The authors may wish to briefly speculate on potential physiological explanations for this trend. For example, could this represent a compensatory growth response to the intermittent, high-volume nutrient delivery characteristic of TRF? Such an adaptation might enhance the absorptive capacity in response to the feeding pattern.
  5. The transcriptomic data provide excellent insights, particularly the upregulation of amino acid transporter genes like SLC38A2 in the jejunum (Lines 423-427). However, gene expression (mRNA) does not always correlate directly with protein levels and functional activity. The authors should acknowledge this limitation in the discussion. Mentioning that future studies using proteomics or Western blotting could validate these transcriptomic findings would strengthen the paper.
  6. Role of Axon Guidance Pathway: The analysis revealed that the 62 DEGs shared between the jejunum and colon were primarily enriched in the axon guidance pathway (Lines 276-279). The authors suggest this might implicate gut-brain interaction. This is a very interesting point that could be slightly elaborated upon. How might TRF, via altered nutrient sensing or microbial metabolites, modulate the enteric nervous system through these pathways? A sentence or two of additional speculation would enhance the discussion.
  7. Clarity on Enzyme Data: The results state that TRF increased activities of certain enzymes in the pancreas and duodenum, citing a previous report (Lines 193-196). For reader convenience and to make this manuscript a more self-contained piece of work, it would be beneficial to include a summary of this key data, perhaps in a supplementary table, rather than only referencing a previous publication.

Author Response

Based on the evaluation of the submitted manuscript titled "Segment-specific functional responses of swine intestine to time-restricted feeding regime," this work addresses a relevant and timely topic in swine production by examining the effects of time-restricted feeding on intestinal function. The discovery that time-restricted feeding affects the jejunum and colon differently, improving protein utilization in the upper intestine and carbohydrate metabolism in the lower intestine, is novel and makes an important contribution to the field. The methods are appropriate overall, and the results are clearly presented. However, some points would benefit from further clarification and slight expansion to enhance the manuscript’s conclusions and overall significance. Please kindly follow the comments and suggestions provided below;

  1. The study indicates elevated pancreatic chymotrypsin and lipase activities alongside reduced amylase, sucrase, and lipase activities in the ileum (Lines 191-196). The authors suggest this pattern reflects an energy conservation approach, given more proximal digestion of substrates (Lines 314-318). Although this explanation seems reasonable, it represents a key interpretation that would gain from additional evidence. Could the authors elaborate on potential signaling pathways underlying this regional enzyme reduction? For example, might changes in nutrient or bile acid flow from time-restricted feeding activate feedback loops that limit enzyme production in the distal small intestine?

Answer: Thank you for your suggestions which have provided us with excellent analytical insights and future research directions. In the current study, we did not delve into the specific mechanisms underlying the changes in these digestive enzymes. However, our transcriptome results did reveal alterations in key genes involved in bile acid metabolism-related pathways, and we have added discussions on this part in the Discussion section. We will also prioritize the investigation of relevant content in subsequent studies. Thank you again for your valuable suggestions.

  1. The manuscript reports a notable 18.9% rise in Apparent Total Tract Digestibility (ATTD) of crude protein (Lines 185-188). Authors then link this to potential boosts in colonic microbial protein synthesis, drawing on nitrogen from endogenous and microbial pools (Lines 352-354). This hypothesis holds interest, yet ATTD for crude protein incorporates microbial nitrogen, which may complicate assessments of actual protein uptake. Did the authors evaluate apparent ileal digestibility (AID) of amino acids? Such analysis would offer a sharper gauge of small intestine protein absorption, separating it from hindgut fermentation influences and bolstering claims that time-restricted feeding improves proximal protein use.

Answer: Thank you for your question. In this study, we primarily investigated the effects of TRF on the ATTD of proximate nutrients. Your suggestions are highly professional, and in future research, we will further explore the impacts of TRF on the apparent (standardized) ileal amino acid digestibility of growing pigs.

  1. The study observes that time-restricted feeding (TRF) lowered colonic cellulose by 18% significantly, yet showed no notable change in Apparent Total Tract Digestibility (ATTD) of crude fiber (CF) (Lines 203-206). The authors appropriately highlight high individual variation and differences in CF versus cellulose measurement methods (Lines 354-359). This discussion merits further development, as the cellulose drop represents a core result tied to shifts in hindgut metabolism, while the absent ATTD-CF effect needs deeper exploration. Could components within CF, such as unmeasured lignin, account for this inconsistency, since cellulose assays exclude it? Addressing this would enrich the interpretation with greater detail.

Answer: Thank you for your question. In accordance with your suggestion, we have included relevant discussions in the Discussion section.

  1. The study reports a trend toward increased jejunum length in the time-restricted feeding (TRF) group (P=0.07, Line 177), which, although not statistically significant, is a notable morphological observation. The authors may wish to briefly speculate on potential physiological explanations for this trend. For example, could this represent a compensatory growth response to the intermittent, high-volume nutrient delivery characteristic of TRF? Such an adaptation might enhance the absorptive capacity in response to the feeding pattern.

Answer: Thank you for your question. Inspired by your inquiry, we plan to conduct intestinal tissue sections to examine indicators such as intestinal villus height, crypt depth, and villus height-to-crypt depth ratio in subsequent studies. These indicators are closely associated with the digestion and absorption of nutrients. For the manuscript itself, we have added discussions related to the relevant data in the Discussion section.

  1. The transcriptomic data provide excellent insights, particularly the upregulation of amino acid transporter genes like SLC38A2 in the jejunum (Lines 423-427). However, gene expression (mRNA) does not always correlate directly with protein levels and functional activity. The authors should acknowledge this limitation in the discussion. Mentioning that future studies using proteomics or Western blotting could validate these transcriptomic findings would strengthen the paper.

Answer: We appreciate your suggestion and have added discussions on the relevant content in the Discussion section.

  1. Role of Axon Guidance Pathway: The analysis revealed that the 62 DEGs shared between the jejunum and colon were primarily enriched in the axon guidance pathway (Lines 276-279). The authors suggest this might implicate gut-brain interaction. This is a very interesting point that could be slightly elaborated upon. How might TRF, via altered nutrient sensing or microbial metabolites, modulate the enteric nervous system through these pathways? A sentence or two of additional speculation would enhance the discussion.

Answer: We appreciate your suggestion and have added discussions on the relevant content in the Discussion section.

  1. Clarity on Enzyme Data: The results state that TRF increased activities of certain enzymes in the pancreas and duodenum, citing a previous report (Lines 193-196). For reader convenience and to make this manuscript a more self-contained piece of work, it would be beneficial to include a summary of this key data, perhaps in a supplementary table, rather than only referencing a previous publication.

Answer: Thank you for your valuable suggestions; we have supplemented the data referring to digestive enzyme activities in the pancreas and duodenum in the Figure S1 and cited in the manuscript.